# The Bone Biomarker of Quantitative Chemical Shift Imaging in Patients with Type 1 Gaucher Disease Receiving Low-Dose Long-Term Enzyme Replacement Therapy

**DOI:** 10.3390/jcm12062220

**Published:** 2023-03-13

**Authors:** Ari Zimran, Jeff Szer, Michal Becker-Cohen, Sjoerd Jens, Claudia Cozma, Shoshana Revel-Vilk

**Affiliations:** 1Gaucher Unit, Shaare Zedek Medical Centre, Jerusalem 9103102, Israel; 2Faculty of Medicine, The Hebrew University of Jerusalem, Jerusalem 91120, Israel; 3Peter MacCallum Cancer Centre, Royal Melbourne Hospital, Melbourne, VIC 3000, Australia; 4Department of Medicine, University of Melbourne, Melbourne, VIC 3052, Australia; 5Amsterdam Medical Center, 1105 Amsterdam, The Netherlands; 6Department of Radiology, Rijnstate Hospital, 6815 Arnhem, The Netherlands; 7Centogene AG, 18055 Rostock, Germany

**Keywords:** bone marrow, enzyme replacement therapy, Gaucher disease, quantitative chemical shift imaging, treatment switch

## Abstract

Quantitative chemical shift imaging (QCSI) is the most sensitive imaging biomarker to assess bone marrow involvement in Gaucher disease. Widespread QCSI use is limited by test availability. Anecdotal reports describe two patients demonstrating significant improvement in fat fraction (FF) assessed by QCSI following a switch from imiglucerase to taliglucerase alfa. This analysis evaluated bone marrow involvement in adults with Type 1 Gaucher disease receiving low-dose enzyme replacement therapy (ERT) with imiglucerase and/or velaglucerase alfa. We report baseline data for 30 patients meeting eligibility criteria. Median (range) duration and dose of ERT were 18 (5–26) years and 30 (30–60) U/kg/month, respectively. Low FF scores (<0.30) were observed for seven patients (23%; 95% confidence interval, 10–42%) and were more common in females (*n* = 6) versus males (*n* = 1; *p* < 0.025); one female was menopausal. These baseline data demonstrate that prolonged low-dose ERT with imiglucerase or velaglucerase alfa led to an adequate bone response, assessed by QCSI, in the majority of patients. A minority of such patients with suboptimal bone response require therapeutic change. The next phase of the study will address the effect of switching to taliglucerase alfa on bone status for patients with less than optimal QCSI scores (<0.30).

## 1. Introduction

Gaucher disease is a common lysosomal storage disorder in which recessive mutations in the β-glucocerebrosidase gene (*GBA1*) lead to a deficiency of lysosomal glucocerebrosidase activity [1]. The resultant accumulation of glucocerebroside in macrophages causes organ and tissue damage in the spleen, liver, bone, and bone marrow [1]. Patients with Type 1 Gaucher disease, the most common form of the disease in Europe, Israel, and the United States, lack primary central nervous system involvement [2].

Bone complications, which include osteopenia, fractures, necrosis, and joint destruction, are the most dramatic and life-impairing consequences of Type 1 Gaucher disease [2,3]. Patients experience bone crises (i.e., acute onset of intense pain requiring immobilization and lasting ≥1 week) and chronic bone pain [3]. Gaucher cell infiltration of the bone marrow substantially decreases the bone marrow fat fraction (FF), and the extent of this reduction correlates with the overall severity of skeletal manifestations [2,3,4]. Previous studies have demonstrated that the degree of Gaucher cell infiltration is best estimated by Dixon magnetic resonance imaging-based quantitative chemical shift imaging (QCSI), which measures the displacement of fatty marrow by Gaucher cells [4]. Findings have shown that the FF score can predict the risk of clinically important bone events [4], and can be useful in evaluating the bone marrow response to treatment [5].

Enzyme replacement therapy (ERT) is effective in patients with Type 1 Gaucher disease. The three US Food and Drug Administration-approved ERTs (i.e., imiglucerase [6], velaglucerase alfa [7], and taliglucerase alfa [8]) have favorable safety profiles and have been shown to improve hematologic parameters and quality of life within a few months, reduce liver and spleen volumes within 2 years, and improve some bone abnormalities within 9 months to 4 years of treatment initiation [9,10,11]. A post hoc analysis of 12 patients with Type 1 Gaucher disease who participated in a phase I/II trial of velaglucerase alfa suggested that improvements in lumbar spine bone marrow burden were maintained through 7 years of ERT in eight patients with data [11]. Although several reports have described the effects of ERT on bone marrow involvement in adult patients with Type 1 Gaucher disease, most long-term data are obtained from small numbers of patients with median treatment durations of <15 years, receiving high-dose regimens, and with methods less quantitative than QCSI [12,13,14,15,16,17].

We sought to establish the effect of long-term ERT on bone marrow more definitively. Thus, the aim of this analysis was to evaluate bone marrow involvement in a larger population of adult patients with Type 1 Gaucher disease who had received low-dose ERT with imiglucerase and/or velaglucerase alfa for a longer period (median of 18 years, and at least 5 years) than previously reported.

## 2. Results

### 2.1. Characteristics of Study Participants

In total, 30 patients (13 females, 17 males) with a median (range) age of 46 (19–71) years consented to participate in this study and undergo QCSI testing. The distribution of *GBA1* mutation types among these patients included N370S homozygotes (c.1226A > G, p.N095) (*n* = 12), p.N370S compound heterozygotes (*n* = 17), and p.T431 homozygotes (c.245C > T p.T82I) (*n* = 1). Eight patients had undergone splenectomy, of which one was a partial splenectomy.

The median (range) duration of ERT was 18 (range 5–26) years. Thirteen patients were receiving imiglucerase as the primary ERT (median [range] duration, 19 (9–26) years), five patients were receiving velaglucerase alfa (median [range] duration, 11 (5–12) years), and twelve patients had switched from imiglucerase to velaglucerase alfa (median [range] duration, 7 (6–10) years). ERT doses were 15 U/kg, 30 U/kg, or 45 U/kg every 2 weeks. Most patients received the low-dose regimen (15 U/kg every 2 weeks) (Table 1) because treatment in Israel is typically initiated at a low dose, with the dose increased in cases of inadequate response.

### 2.2. Study Outcome Measures

The median (range) lumbar spine T score was −1.3 (−2.8 to 0.0) among the 26 patients for whom dual energy X-ray absorptiometry (DEXA) scans were available. Two patients had a score in the osteoporosis range (<−2.5) and 11 had a score in the osteopenia range (2.5 to −1.0). No patient had clinical bone disease or had experienced a catastrophic bone event. The median (range) QCSI FF scores were 0.42 (0.24–0.66) among all 30 patients, 0.39 (0.24–0.66) among females, and 0.44 (0.28–0.60) among males. Abnormal QCSI FF scores (<0.30) were found for seven (23%) patients (95% confidence interval [CI], 10–42%). Abnormal QCSI FF scores were more common in females than in males (Table 1; *p* = 0.025), adjusted for age. The QCSI for the p.T431 homozygote (0.59) was within the normal range, and there were no notable observations for this patient. No differences in age, genotype, history of splenectomy, duration and type of ERT, and disease-related parameters were found between patients with a QCSI score of bone at risk and those with a normal QCSI score (Table 1). In addition, no differences were observed in lysosomal glucosylsphingosine-1 (lyso-Gb1) for patients with reduced FF versus the remaining 75% patients with normal QCSI scores.

## 3. Discussion

We used QCSI, a method considered to provide the most sensitive and accurate quantitative assessment of bone marrow involvement in Type 1 Gaucher disease [18], to study bone responses in 30 patients with Type 1 Gaucher disease who had received ERT for a median of 18 years, and at least 5 years. The majority of patients who received prolonged low-dose ERT with imiglucerase and/or velaglucerase alfa showed a normal QCSI score (bone marrow infiltration in a “safe” range), indicating positive bone status. While QCSI low FF values are considered predictive of future skeletal complications, the lack of lyso-Gb1 values > 147 ng/mL in this study’s patient population, which excludes children and patients with Type 3 Gaucher disease, reflects the actual lack of such complications among the seven patients with low FF; thus, it may be acceptable to rely on this biomarker given the lack of access to QCSI globally. Because of the required technical expertise and the cost of the instrumentation, the use of QCSI was limited to academic medical centers for many years [18,19]. However, QCSI may gain wider use in routine clinical practice as imaging vendors release SE Dixon techniques in their scanner sequences [18,20].

Approximately 20% of patients had QCSI scores indicative of a suboptimal bone response (six females, one male). A low bone marrow FF was recorded mainly in premenopausal women. The observation that a greater number of females than males showed a suboptimal bone response may potentially relate to the hormonal status in women of reproductive age; however, as no studies have specifically evaluated the association between FF and hormonal status, we are not able to explain the differences observed. As no other patient- or disease-related parameter predicted abnormal bone marrow infiltration, a more widely available, quantitative measure of bone marrow infiltration may be required for the assessment of the bone response to ERT for patients with Type 1 Gaucher disease. Assessment of bone mineral density may “overcall” the effect of Gaucher disease on bones and may not be a reliable guide for determining the efficacy of ERT.

A study of patients from the International Collaborative Gaucher Group (ICGG) Gaucher Registry who had bone crises and/or bone pain for 1 year before starting ERT and yearly for 3 years after the start of ERT (dose of ERT not reported) found that the incidence of bone crises decreased from 17% the year before ERT to 5%, <1%, and 3% at 1, 2, and 3 years, respectively, after the start of ERT [21]. For bone pain, the incidence decreased from 49% the year before ERT to 30%, 29%, and 30% at 1, 2, and 3 years, respectively, after the start of ERT [21]. No bone complications were reported among our patients during the course of long-term ERT.

The potential effect of a switch to another ERT product on bone status will be determined in the next phase of this study for the patients with less than optimal QCSI scores (<0.30). These patients will be offered treatment with taliglucerase alfa at equivalent doses and will undergo subsequent reassessment of any impact of treatment on clinical symptoms and QCSI scores after 1 year and 2 years. An exploratory analysis of QCSI scores in eight treatment-naive patients participating in a phase III study of taliglucerase alfa suggested that treatment increased the bone marrow FF compared with baseline and compared with a cohort of 15 untreated patients after 1 year and up to 3 years of follow-up [10]. In addition, improvements in QCSI scores were observed in an analysis of the effects of taliglucerase alfa in treatment-naive patients as well as in patients who switched from imiglucerase treatment, suggesting the potential for taliglucerase alfa to improve responses to prior ERT [22].

We found that the median FF was numerically higher among men than women (0.44 versus 0.39). In a study of lumbar vertebral marrow fat distribution in 46 healthy adults (women, *n* = 21; men, *n* = 25; aged 18–77 years), FF was significantly higher for men than for women (mean [SD], 39.5% [12.2] vs. 35.9% [13.9]; *p* < 0.001) and for postmenopausal women than for premenopausal women (mean [SD], 44.5% [17.0] vs. 29.6% [7.8]; *p* < 0.001) using univariate regression [23]. However, only age had a significant effect on FF by vertebral level in multivariate regression analysis using a robust variance estimator (*p* = 0.003; β = 0.52; standard deviation 0.18). Other independent variables (body mass index, weight, and height) had no significant effect on either univariate or multivariate analysis [23].

Limitations of the current study include the lack of a comparator (baseline or untreated historical controls), assessment at only one time point, and conduct in patients from a single site. Its strengths are its long treatment duration, as patients had received ERT for at least 5 years and for up to 26 years, longer than in other published studies, and its use of QCSI, as this method has not been used to assess such long-term effects of ERT on bone.

## 4. Materials and Methods

### 4.1. Study Population

Eligible patients were ≥18 years of age with proven Type 1 Gaucher disease, based on clinical evaluation, documentation of deficient enzyme activity, and genotyping, who were treated with ERT (imiglucerase or velaglucerase alfa) at the Shaare Zedek Medical Centre (SZMC; Jerusalem, Israel) Gaucher Unit for ≥5 years, had received a stable dose of ERT in the previous 6 months, and were willing to travel to the Academic Medical Center (AMC), Amsterdam, the Netherlands. Patients taking another experimental drug or with past exposure to taliglucerase alfa, as well as patients with any medical, emotional, behavioral, or psychological condition, were excluded. Written informed consent was provided by patients before participation in any study procedure.

### 4.2. Study Design

In this investigator-initiated clinical study for switch of patients with low QCSI scores, patient clinical, laboratory, and imaging data were collected from patient charts for the last visit prior to QSCI. Dried blood spots were collected on filter cards (CentoCard^®^, Centogene AG, Rostock, Germany), and lyso-Gb1 analysis was carried out at Centogene. The lyso-Gb1 levels were measured using liquid chromatography-tandem mass spectrometry of dried blood spot samples (Centogene AG, Rostock, Germany) as previously described [24,25]. According to the Centogene Laboratory, the normal range for lyso-Gb1 is <6.8 ng/mL [24].

Lumbar spine (L1–L4) T scores were determined by DEXA performed at SZMC, as reported previously [26], and bone marrow involvement was assessed by Dixon QCSI performed in the Department of Radiology at the AMC based on an established method [4]. Dixon QCSI uses the mean of the FF in vertebrae L3–L5 to measure the infiltration of bone marrow by Gaucher cells [4,5]. A QCSI score < 0.30 was indicative of a bone at risk. The mean of the readings for vertebrae L3–L5 was used in an algorithm developed by the AMC to calculate lumbar spine FF [22]. Consecutive measurements were standardized by repositioning the image slide on the mid-sagittal localizer as closely and consistently as possible [22].

### 4.3. Statistical Analysis

The study sample size was based on budget limitations. Based on our previous study of bone marrow Gaucher cell infiltration in patients with Type 1 Gaucher disease treated with taliglucerase alfa [22], we assumed that screening 30 patients would enable the identification of approximately 20 patients with abnormal QCSI scores.

Data were summarized with descriptive statistics. Median (range) and 95% CIs were determined for continuous variables. Categorical variables were summarized as the number of patients and percentage of patients. Logistic regression was done to evaluate predictors of abnormal QCSI FF (<0.3). A *p*-value < 0.05 was considered significant. All data analyses were carried out with SPSS software (statistical package version 25.0; IBM, Armonk, NY, USA).

## 5. Conclusions

Most patients with Type 1 Gaucher disease receiving long-term ERT with imiglucerase and/or velaglucerase alfa have bones that are not at risk for irreversible events, as assessed by QCSI. Bone mineral density assessment may not accurately reflect the impact of Gaucher disease on bones, and thus, it is likely not a reliable guide to the efficacy of ERT on bones. A more widely available quantitative measure of bone marrow infiltration is needed for assessment of response to therapies for individuals with Type 1 Gaucher disease, and the impact of alternative ERT on suboptimal responders is to be determined.

## Figures and Tables

**Table 1 jcm-12-02220-t001:** Characteristics of patients with low QCSI scores (<0.3) and normal QCSI scores.

Characteristics	QCSI FF Score < 0.3 *n* = 7	QCSI FF Score ≥ 0.3 *n* = 23	*p* Value *
Age, median (range), years	46 (23–71)	48 (19–60)	0.105
Female, *n* (%)	6 (85.7)	7 (30.4)	0.015
p.N370S homozygote, *n* (%)	4 (57)	8 (35)	0.86
Splenectomy, *n* (%)	2 (28.5)	6 ^†^ (26.1)	0.25
Platelet count × 10^9^/L, median (range)	156 (141–268)	172 (67–402)	0.89
Hemoglobin concentration in g/L, median (range)	130 (111–150)	152 (111–172)	0.39
Lysosomal-Gb1 in ng/mL, median (range)	111 (68.7–147)	77.9 (11.9–304)	0.35
T score lumbar spine, median (range)	−1.2 (−2.1 to −0.5)	−1.3 (−3 to 0.6)	0.94
Time on ERT in years, median (range)	21.0 (8.5–25)	16.5 (5–26)	0.06
ERT dosage, *n*			0.14
15 U/kg every 2 weeks	5	16	
30 U/kg every 2 weeks	2	5	
45 U/kg every 2 weeks	0	2	

* Logistic regression analysis. ^†^ One patient had a partial splenectomy. ERT, enzyme replacement therapy; FF, fat fraction; Gb1, glucosylsphingosine-1; QCSI, quantitative chemical shift imaging.

## Data Availability

Data cannot be shared due to ethical and privacy issues.

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
