# Peer review of "The Bone Biomarker of Quantitative Chemical Shift Imaging in Patients with Type 1 Gaucher Disease Receiving Low-Dose Long-Term Enzyme Replacement Therapy"

_jcm, 2023, doi:10.3390/jcm12062220_

Round 1

Reviewer 1 Report

In this manuscript, Zimran et al. have used Quantitative chemical shift imaging (QCSI) to evaluate the effect of long-term, low-dose ERT (imiglucerase and/or velaglucerase alfa) on bone marrow fat fraction (FF).  These sensitive measurements are used to evaluate bone responses, which are more refractory to ERT.  The authors evaluated 30 patients with type 1 GD meeting eligibility criteria.  The data obtained showed that prolonged low-dose ERT with these enzyme preparations led to an adequate bone response in the majority of patients.  The study was well planned and executed, and the data obtained will be very valuable to physicians and clinical investigators.

Reviewer 2 Report

I was very pleased to review the article „The Bone Biomarker of Quantitative Chemical Shift Imaging in Patients with Type 1 Gaucher Disease Receiving Low-Dose Long-term Enzyme Replacement Therapy“. The study was well-designed and the authors clearly stated its limitations. Results emphasize the value of QCSI as a sensitive imaging method for assessing bone involvement in Gaucher disease and response to the treatment.

I have two remarks:

1.      Since QCSI FF scores are the main results presented in this study it is necessary to describe the method. It is not enough to cite a publication from 2002.

2.      There are publications showing that females have lower FF than males in the population without bone disease. One of them is Ognard J, Demany N, Mesrar J, Aho-Glélé LS, Saraux A, Ben Salem D. Mapping the medullar adiposity of lumbar spine in MRI: A feasibility study. Heliyon. 2021 Jan 16;7(1):e05992. Please consider commenting on that in the discussion section. I would also advise adding data on median QCSI FF scores presented separately for male and female patients. 

Reviewer 3 Report

Thank you for including me in the review of this article on bone biomarker assessment in patients with type 1 Gaucher disease receiving low-dose long-term treatment with enzyme replacement therapy. The overall recommendation is to accept the article in its present state, minor revisions requested. While there are inherent limitations to the study, these were clearly presented and described and overall, the findings reported should be of great interest to the scientific community, especially those who are directly involved in the management and care of patients with Gaucher disease.

Brief summary:

In this article, Zimran et al describe their findings on quantitative chemical shift imaging (QCSI) as a bone imaging biomarker to assess bone marrow involvement in 30 patients with type 1 Gaucher disease on long-term treatment with ERT. These findings are valuable to the field and as more studies are done to support these findings, QCSI analysis could be a helpful tool to assess bone involvement. The follow-up on switch patients and potiential follow-up on the significance in post-menopausal women to provide supporting data to see if there is an association with hormones will be of interest to the scientific community.

Recommendations: The authors state that the QCSI is available globally and may be a more acceptable biomarker for bone involvement, can the authors expand upon the QCSI in the methodology section? Provide more commentary regarding the use of QCSI in assessing bone lesions per the literature.

Update the N370S to most current nomenclature and use the p. in front of all genotypes provided in the text and table.

Can you provide the cDNA and full protein change for the T431 homozygote listed on line 73; and as there is only one patient with this genotype, recommend to change homozygotes to homozygote on line 73.

Recommend to remove the parenthesis around on line 73 for the protein change p.(Asn409Ser), also recommend to have this in the same one letter abbreviation as the p.N370S, or have them both be the three letter abbreviations to be consistent throughout the manuscript.

Is a normal reference range provided for the lyso-Gb1 reports? If so, recommend to include this in the study design paragraph referencing the analysis of the analyte.

What was the QCSI score for the p.T431 homozygote? Did anything stand out with this one patient that the authors can comment upon in the discussion?

In the switch patients from imiglucerase to taliglucerase alfa, did anything stand out as far as clinical improvements with other measured analytes (hemoglobin, platelets, lyso-Gb1)? The table doesn’t distinguish these but it is mentioned in the text.
